# Concentration- and Time-Dependent Dietary Exposure to Graphene Oxide and Silver Nanoparticles: Effects on Food Consumption and Assimilation, Digestive Enzyme Activities, and Body Mass in *Acheta domesticus*

**DOI:** 10.3390/insects15020089

**Published:** 2024-01-29

**Authors:** Reyhaneh Seyed Alian, Barbara Flasz, Andrzej Kędziorski, Łukasz Majchrzycki, Maria Augustyniak

**Affiliations:** 1Institute of Biology, Biotechnology and Environmental Protection, University of Silesia in Katowice, 40-007 Katowice, Poland; reyhaneh.seyed.alian@us.edu.pl (R.S.A.); barbara.flasz@us.edu.pl (B.F.); andrzej.kedziorski@us.edu.pl (A.K.); 2Institute of Physics, Faculty of Materials Engineering and Technical Physics, Poznan University of Technology, Piotrowo 3, 60-965 Poznan, Poland; lukasz.majchrzycki@put.poznan.pl

**Keywords:** house crickets, food consumption and assimilation, protease, amylase, α-glucosidase, β-glucosidase, β-galactosidase, lipase

## Abstract

**Simple Summary:**

The increasing presence or contamination of various everyday-life products (including foodstuffs) by graphene oxides and silver nanoparticles (GOs and AgNPs, respectively) raises a risk of their possibly deleterious effects on digestive functions and, consequently, nutrient and energy intake by an organism. The study addresses this issue by considering various NP concentrations and exposure times. The scarcity of relevant data makes such studies necessary for the reliable assessment of NP effects. This study on a model insect species—adult house crickets—revealed a changed profile of digestive enzymes’ activities in the gut, mainly when a high content of NPs was present in the food: stimulated digestion of carbohydrates and lipids but inhibited digestion of proteins. These changes were more pronounced in AgNP-treated than in GO-treated insects and increased with exposure time. Disturbed digestion led to decreased food consumption with exposure time in AgNP-treated crickets. Food assimilation was also affected—the cumulative food assimilation (CFA) was higher and lower compared with the control in crickets exposed to the lowest and moderate concentrations of AgNPs, respectively. These findings confirmed weak or no effects of low amounts of NPs in food and revealed that their higher concentrations may adversely influence digestive processes and resulting nutrient and energy intakes, particularly during prolonged exposure of an organism.

**Abstract:**

The advancement of nanotechnology poses a real risk of insect exposure to nanoparticles (NPs) that can enter the digestive system through contaminated food or nanopesticides. This study examines whether the exposure of model insect species—*Acheta domesticus*—to increasing graphene oxide (GO) and silver nanoparticle (AgNP) concentrations (2, 20, and 200 ppm and 4, 40, and 400 ppm, respectively) could change its digestive functions: enzymes’ activities, food consumption, and assimilation. We noticed more pronounced alterations following exposure to AgNPs than to GO. They included increased activity of α-amylase, α-glucosidase, and lipase but inhibited protease activity. Prolonged exposure to higher concentrations of AgNPs resulted in a significantly decreased food consumption and changed assimilation compared with the control in adult crickets. A increase in body weight was observed in the insects from the Ag4 group and a decrease in body weight or no effects were observed in crickets from the Ag40 and Ag400 groups (i.e., 4, 40, or 400 ppm of AgNPs, respectively), suggesting that even a moderate disturbance in nutrient and energy availability may affect the body weight of an organism and its overall condition. This study underscores the intricate interplay between NPs and digestive enzymes, emphasizing the need for further investigation to comprehend the underlying mechanisms and consequences of these interactions.

## 1. Introduction

The rapid advancement of nanotechnology poses an increasing challenge to numerous species in diverse ecosystems due to exposure to various nanoparticles (NPs). Although the severity of ecosystem infiltration by NPs remains uncertain, one may expect their increasing concentrations due to growing applications in various fields [1,2,3,4].

Many NPs, including graphene oxide (GO) and silver nanoparticles (AgNPs), have recently drawn attention as undesirable food contaminants [5,6,7,8,9]. Contemporary applications of GO in the cultivation of edible plants, food production, processing, and packaging, in order to prevent bacterial growth and spoilage, may increase its content in a variety of food products. Other possible sources of their contamination with NPs include biosensors, food quality detectors, food composition analyzers, and NP-containing disinfectants and coatings. Most studies highlighting innovative applications of NPs in the food industry emphasize the need for comprehensive toxicological research to understand their effects on digestive functions and, consequently, the organisms’ conditions [8,9,10,11,12].

Nanoparticles can influence food digestion and, consequently, impact nutrient availability and assimilation, which may disturb an organism’s development and performance [13,14,15,16,17]. Exposed animals may respond to the stressor by increasing food intake to maintain the efficiency of food utilization or by accelerating food passage through the digestive tract, leading to decreased assimilation [18]. However, under certain conditions, compensatory mechanisms may be activated [19].

There are few in vivo studies on NP effects on digestive enzymes, particularly in invertebrates, and a lack of comprehensive ones that encompass various NP concentrations and exposure durations. Some authors noticed inhibition of the enzymes’ activity [13,15,20,21,22,23], but we observed, in a semi-quantitative screening approach, increased gut enzyme activity in insects exposed to low concentrations of AgNPs or GO [19]. Therefore, thorough investigations are required to explain these contradictory results, considering a broad range of both NP concentrations and exposure periods.

In this study, we compared the effects on the digestive functions of two types of nanoparticles (AgNPs and GO) that may display somewhat different modes of action. The proposed mechanisms of AgNP action include interaction with the cell membrane and its penetration, interactions with proteins and other cellular components, and enhancement of oxidative stress. The gradual release of silver ions from the NPs’ surface can elicit time-delayed and prolonged effects [24,25,26,27,28]. Graphene oxide, an allotrope of carbon occurring in variously sized flakes, primarily affects target molecular structures by provoking oxidative stress [29,30,31]. Adhesion of GO particles to the cell membrane is also suggested [32]. Such adhesion to the surface of the gut epithelium may impair its functions, even without the internal penetration of GO.

The main aim of the study was to determine which of the selected nanoparticles and what concentration of them could change the activity of digestive enzymes and to establish if there was a causal relationship among the type of nanoparticles, their concentration, exposure time, the activity of selected digestive enzymes, and the amount of food consumption. We focused on detailed, quantitative measurements of the activity of selected digestive enzymes in *Acheta domesticus* exposed to nanoparticles at various concentrations. Furthermore, we performed the measurements over an extended period of time, covering almost the entire life span of the adult insect.

Our research was conducted on *Acheta domesticus* (Gryllidae, Orthoptera), a model organism in physiological and toxicological research that offers numerous advantages for designing in vivo experiments [33]. This species originates from southwestern Asia but is currently distributed worldwide. These insects have well-understood biology, a relatively short life cycle, and a sufficiently large size for research purposes [34,35]. Recently, it has gained much attention as a possibly edible species, raising hopes for addressing global protein food shortages [36,37,38,39].

## 2. Materials and Methods

### 2.1. NP Characteristics

Silver nanoparticles (99.9%, 20–30 nm) were purchased from SS Nanomaterials, Inc. (Houston, TX, USA) and prepared as a stable colloidal stock solution (10 mg/L) by sonication (UP-100H, DONSERV, Warsaw, Poland; cycle 1, amplitude 100%) in a citrate buffer solution (0.1 M; 10 mL; pH = 6.5; 30 min). Graphene oxide (GO) was purchased from Nanografi (Ankara, Turkey) and supplied as an aqueous suspension (10 mg/mL). Before use, the material was appropriately diluted, sonicated, and subjected to morpho-structural and physical analysis. Nanoparticles (NPs) were visualized and measured using microscopy techniques: scanning electron microscopy (SEM; Quanta FEG 250, FEI, Hillsboro, OR, USA) and atomic force microscopy (AFM) (Agilent 5500) (Agilent Technologies, Santa Clara, CA, USA). The electrokinetic potential (zeta potential) of the NP water in aqueous suspension was measured at 25 °C using a Litesizer 500 (Anton Paar, Graz, Austria). A detailed description of the sample preparation procedure for the analysis was published previously [19,29].

The silver nanoparticles had a size of ca. 20 nm and tended to form aggregates 50–60 nm in diameter (Figure 1A,B). The zeta potential was −44.5 mV, confirming the good stability of the suspension (Figure 1C). The microscopic analysis of the GO suspension confirmed the presence of flakes, mainly single-layered with a diameter of up to several µm (Figure 1A’,B’). The zeta potential (−30.2 mV) indicated the good stability of the GO suspension used for food preparation (Figure 1C’).

### 2.2. Acheta Domesticus

The insects used in the experiment were obtained from our stock colony, maintained at the Institute of Biology, Biotechnology, and Environmental Protection at the University of Silesia in Katowice for over 30 years [40]. The breeding conditions were monitored and controlled to maintain optimal ranges of temperature (28.6 ± 1 °C), a photoperiod (L:D 12:12), and humidity (35–47%). The insects were fed the standard pellet food for rabbits (KDT; UNIPASZ, Siemiatycze, Poland; see Appendix A for detailed description), and in the experiment, they were provided with food containing various concentrations of graphene oxide (GO) or silver nanoparticles (AgNPs).

### 2.3. Experimental Design

Cricket eggs were obtained from the laboratory stock colony and kept on a wet substrate. The hatched nymphs were reared in a plastic fauna box under standard conditions until they reached the imago stage. The 1-day-old adults were randomly assigned to seven experimental groups: a control group and groups fed with the feed with GO or AgNPs admixtures, each at three different concentrations (Figure 1). The food was prepared following the protocol established in our laboratory and described previously [19,41]. Briefly, ground rabbit pellets were mixed with the appropriate GO or AgNP suspension volume in distilled water. The final concentrations of GO or AgNPs were as follows: 2, 20, and 200 μg GO/g of food (experimental groups GO2, GO20, and GO200, respectively) or 4, 40, and 400 μg AgNPs/g of food (experimental groups Ag4, Ag40, and Ag400, respectively). The feed for the control group was prepared the same way but with distilled water added instead of the nanoparticle suspension. Then, the feed was dried for 48 h in a dryer (Pol-Eko Aparatura, Wodzislaw Slaski, Poland) at 45 °C and sterilized for 48 h in a laminar chamber (UVcleaner, BIOSAN, Warren, MI, USA). The insects were housed in plastic fauna boxes (28 cm × 20 cm × 16 cm) under standard conditions with unrestricted food, water, and shelter access throughout the experiment.

### 2.4. Food Consumption and Assimilation

The assessment of food consumption and assimilation was conducted following the procedure described earlier [19]. Each experimental group comprised six replicates, with six individuals in each (36 individuals per treatment and 252 insects in the whole experiment). The mass of the provided food, food residues, feces, and insects was accurately measured at 2-day intervals during the first ten days and at 5-day intervals subsequently (Figure 1). Feed and feces samples were weighed after drying (50 °C for 48 h) with 1 mg accuracy (Semi-Micro Balance EX225D, OHAUS, Parsippany, NJ, USA). From these raw data, food consumption and assimilation (amount of ingested and digested feed) were calculated in mg dry weight per day for each individual and for each time interval [19].

### 2.5. Gut Cell Status

On days 3, 5, and 21, the percentage of dead cells in the *A. domesticus* gut was examined. Seven groups (one for each treatment) containing fifteen 1-day-old adults were set up and reared as described above. Five insects from each group were collected on the 3rd, 5th, and 21st days and gently anesthetized on ice. Then, the gut was isolated in 0.1 PBS buffer (pH 7.4; 400 μL; 4 °C). Subsequently, the tissue was gently homogenized (Minilys^®^, Bertin Technologies, Montigny-le-Bretonneux, France), and the resulting cell suspension was used to determine the percentage of dead cells by using flow cytometry (MUSE^®^ Cell Analyzer, Millipore, Billerica, MA, USA), employing the Muse^®^ Annexin V & Dead Cell Kit and following the provided kit protocol. Dead cells were identified as 7-AAD (Aminoactinomycin D—a fluorescent intercalator) positive.

### 2.6. Digestive Enzyme Measurements

For the measurements of digestive enzymes’ activity (protease, amylase, α-glucosidase, β-glucosidase, β-galactosidase, and lipase), gut samples were collected on days 1, 2, 3, 4, 5, 16, and 21 from the start of NP treatment (Figure 1). Samples were obtained from anesthetized crickets by isolating the midgut and homogenizing the tissue in a phosphate buffer (pH 7.4; 1 mL; 4 °C), followed by centrifugation of the homogenates at 14,000 rpm for 10 minutes at 4 °C. Each sample consisted of midguts isolated from three individuals (100 ± 20 mg). Five homogenates were prepared for each experimental group. All analyses of digestive enzymes’ activity were conducted using commercially available kits. Our team has optimized all protocols provided by the manufacturers for *Acheta domesticus* tissues.

To estimate protease activity, the Protease Assay Kit (Calbiochem; Merck KGaA, Darmstadt, Germany; Cat. No. 539125; LOT 3802816) was used, and the activity was measured spectrophotometrically as changes in absorbance at 492 nm per minute. Amylase activity was measured using the Amylase Activity Assay Kit (Sigma-Aldrich, St. Louis, MO, USA; Cat. No. MAK009; LOT 8E24K07110) and expressed in µmol of product/min/mL supernatant. The α-Glucosidase Activity Assay Kit (Sigma-Aldrich, St. Louis, MO, USA; Cat. No. MAK123; LOT 123CA05A04) and β-Glucosidase Activity Assay Kit (Sigma-Aldrich, St. Louis, MO, USA; Cat. No. MAK129; LOT 129CB08A18) were applied to determine α- and β-glucosidase activity, respectively. These assays were based on the substrate-specific product formation reaction rate and were measured spectrophotometrically (TECAN Infinite M200, Männedorf, Austria) in 96-well flat-bottom plates at a wavelength of 405 nm. The activities of α-glucosidase and β-glucosidase were expressed in units/L, with 1 unit equaling the amount of the enzyme that catalyzes the hydrolysis of 1 µmol of substrate/min. The activity of β-galactosidase was assessed using the β-Galactosidase Activity Assay Kit (Abcam, Cambridge, CB2 0AX, UK; Cat. No. ab287846; LOT GR3429797-1). The reaction rate was measured spectrofluorimetrically (HITACHI F-7000 Fluorescence Spectrometer Plate Reader, Tokyo, Japan; Ex/Em = 480/530 nm) for 30 minutes. β-Gal activity was expressed in units/L, with 1 unit representing the amount of enzyme that generates 1.0 μmol of Fluorescein per minute. Lipase activity was measured using the Lipase Activity Assay Kit (Sigma-Aldrich, St. Louis, MO, USA; Cat. No. MAK046; LOT 8H15K07220). The formation rate of the reaction product catalyzed by lipase was measured at 570 nm with glycerol as the standard. Enzyme activity was expressed in µmol/min/mL.

### 2.7. Statistical Analysis

The analyses of digestive enzyme activity were conducted in five replicates, while the food budget analyses included six replicates. Before proceeding with statistical analyses, the assumptions of the analysis of variance were checked. The Kolmogorov–Smirnov and Lilliefors tests were used to assess the data distribution. The Levene and Brown–Forsythe tests were applied to evaluate the homogeneity of variances. For parameters meeting the assumptions of ANOVA (body mass, food consumption, and assimilation, as well as cumulative food consumption and assimilation), the main effects and interactions were analyzed using ANOVA/MANOVA. Then, the differences were examined with a post hoc LSD test. For other parameters, due to the ambiguity of the results of these tests for ANOVA assumption, two statistical approaches were used in the subsequent analysis of the results. To assess the main effects and their interactions, PERMANOVA, a non-parametric test, and multivariate repeated measures ANOVA (Tukey test, *p* < 0.05) were performed. The results of both tests were compared. The charts present results as mean ± SE or median ± interquartile range. Statistical analyses were conducted using STATISTICA^®^ 13.3 (StatSoft Inc., Tulsa, OK, USA) and R (Adonis function, vegan package).

## 3. Results

### 3.1. Body Mass and Food Budget—Consumption and Assimilation

The average body mass gain of insects in the control group after 5 and 21 days of the experiment was 53.6 and 73.3 mg/individual, respectively, indicating a major increase during the first few days of the adult stage (Figure 2. The ANOVA/MANOVA analysis showed a significant effect of “NP type” and “concentration,” as well as both factors’ interactions (Table 1). Post hoc analyses revealed that these effects were mainly related to AgNPs (Figure 2). Treatment with GO did not affect the body weight in *A. domesticus*, except for the GO200 group, where significantly lower weight gain, compared with the control, during 1–5 days of adult life was observed (Figure 2A). The effect of AgNPs depended more on the concentration than on the duration of exposure—the lowest concentrations (4 μg AgNPs/g of food) resulted in the highest weight gain among all groups. The intermediate one (40 μg AgNPs/g of food) slowed down the insect’s growth (Figure 2B). Despite the slowest growth, increased mortality was also observed in the latter group during the second half of the experiment.

Food consumption was measured in 2- and 5-day intervals depending on the expected consumption rate by the crickets and calculated per day per individual for each interval (Figure 3). The effects of “NP type”, their “concentration”, and duration of exposure, as well as their interactions, were analyzed using repeated measures ANOVA (Table 2) and PERMANOVA (Appendix A). The highest consumption occurred during the first 5 days of the imago life in all the groups. The control crickets consumed on average 37.0 and 42.0 mg of dry food per individual per day in the first and second measured intervals, respectively. In the next intervals, consumption in this group decreased and remained in the range of 21.0–25.7 mg per individual per day. The multivariate repeated measures ANOVA analysis revealed that the “NP type” had no significant effect on food consumption. However, the “concentration” and “time” factors significantly influenced it. Interactions between the exposure time and the “NP type” or “concentration” were also significant but not the interaction among all three factors (Table 2), showing that both nanoparticles similarly affected food consumption. However, unlike GO-treated crickets, decreased food consumption in the Ag40 group compared with the control was observed over the entire experiment and in the other AgNP groups in its second half (Figure 3). Detailed results of the post hoc test are provided in Appendix A.

Food assimilation in the control group followed changes in the consumption rate and was the highest during the first 5 days of the experiment, reaching 47.6% of the consumed food. Then, it slowly decreased to 33% in the last measured age interval. All factors included in this experimental model significantly affected food assimilation, with the strongest effect exerted by the “time” factor (Table 2). The main effects analyses revealed significantly higher assimilation in the GO groups treated with the lowest concentration of nanoparticles (55% of the consumed food). Assimilation decreased significantly with the age of the insects but still allowed them to absorb about 50% of the consumed food. A higher assimilation rate than the control was also observed in the AgNP-treated groups, particularly in the Ag4 group. Factor interactions were significant, except for the “time” × “NP type” × “concentration” interaction, suggesting a similar pattern of effects induced by different concentrations of GO and AgNPs throughout the experiment (Figure 4). However, the non-parametric PERMANOVA test showed that all factors and their interactions affected food assimilation (Appendix A). Detailed results of the post hoc analyses for the food assimilation are included in Appendix A.

The obtained results were consistent with the cumulative food budget parameters—cumulative food consumption (CFC) and cumulative food absorption (CFA)—analyzed on day 21 (Table 3). The type of NPs significantly affected CFA, but their concentration affected both parameters. Again, interactions between the “NP type” and “concentration” appeared significant in the combined analysis and the CFA case (Table 3). Post hoc analysis showed that only in the Ag40 group was there a significantly lower CFC. GO caused a significant increase in CFA only in the GO200 group, while AgNPs caused a significant increase in CFA in the Ag4 group and its decrease in the Ag40 group, compared with the control (Figure 5). A visualization of the averaged CFC and CFA changes throughout the experiment is provided in the Appendix A.

### 3.2. Gut Cell Status

The gut cell viability significantly depended on the insects’ age (Figure 6). In the control group, the percentage of dead cells was below 1% at the beginning of the experiment, 3.98% on average on the 5th day, and 20.72% on the 21st day. Compared with the control, nanoparticles did not change the percentage of dead cells on the third and fifth days of the experiment. On the final day of the experiment, crickets from groups GO2 and GO20 maintained a low percentage of dead cells, significantly below that of the control (Figure 6).

### 3.3. Enzyme Activity

Due to the ambiguous results of the tests that verified the assumptions for the analysis of variance, we performed a PERMANOVA analysis, as well as a repeated measures ANOVA (for the latter, see Appendix A), to reveal the effects of experimental factors and their interactions on the activity of digestive enzymes. Both particular factors and their interactions significantly affected the activity of digestive enzymes (Table 4). The analysis of variance with repeated measurements yielded similar results (Appendix A).

The α-amylase activity typical for adult *A. domesticus* in the control group showed slight fluctuations throughout the experiment, with median values ranging between 3.15 and 6.17 µmol/min/mL. In this group, the highest activity was observed in insects on the second day of the experiment, while the lowest was recorded on the fifth day. GO significantly increased α-amylase activity but only in insects from the GO200 group. Lower GO concentrations did not alter α-amylase activity compared with the control, which was confirmed by both parametric and non-parametric tests (Figure 7A and Appendix A). Treatment with AgNPs significantly increased α-amylase activity in the insects from the Ag400 group, where the activity was about 22–44 times that of the Ag400 group compared with the control (Figure 7B and Appendix A).

The activity of α-glucosidase (α-Glu) in the control group increased slowly with the crickets’ age. Both “NP type” and “concentration” had significant effects and indicated a stronger influence of AgNPs than GO (Table 4). The most pronounced increase in α-Glu activity was observed in the Ag4 group. However, on the last day of the experiment, the activity of α-Glu was significantly higher in the Ag40 and Ag400 groups than in the control (Figure 8B and Appendix A). Following GO treatment, the activity of this enzyme did not change, except for single time points for the used concentrations (Figure 8A and Appendix A).

In contrast to α-glucosidase, the basic activity of β-glucosidase (β-Glu) in the control group tended to decrease over time (Appendix A). A significant effect on its activity was stated for both the “NP type” and “concentration” factors (Table 4 and Appendix A). GO groups demonstrated higher β-Glu activity than the control, with the highest values observed in the GO20 group. AgNPs at concentrations of 4 and 400 μg/g of food stimulated the activity of β-Glu. In contrast, an intermediate concentration (40 μg/g of food) inhibited this enzyme (Figure 9). “Time” and its interactions with other factors had a significant but weaker effect on β-Glu activity in comparison with “NP type” and “concentration” and their interaction. This was due to a distinct “enzymatic response” to various concentrations of GO or AgNPs, manifested in groups treated with intermediate concentrations (GO20, stimulation of the enzyme, and Ag40, inhibition of enzyme activity). The significance of the main effects and their interactions was confirmed in both analyses (Table 4, Appendix A and Figure 9 and Appendix A).

The PERMANOVA analysis revealed that all factors and interactions significantly affected β-galactosidase (β-Gal) activity (Table 4). The most substantial effect was attributed to the “NP type” factor and was manifested in higher overall β-Gal activity in the groups exposed to GO compared with insects treated with AgNPs. The main effect of “concentration” was also significant, albeit modified by other factors in the experiment. In the GO groups, the highest overall β-Gal activity occurred at the highest concentration. AgNPs stimulated β-Gal activity at the lowest and inhibited it at the highest concentration (Figure 10 and Appendix A). The main effect of the “time”, analyzed for both nanoparticles and all concentrations simultaneously, indicated an increase in β-Gal activity from the fifth day. “Time” interactions with the other factors were significant, indicating different patterns of changes over time for both nanoparticles and all analyzed concentrations (Figure 10 and Appendix A and Table 4, Appendix A). Figure 10 illustrates only significant differences between the NP-treated groups and the control at each time point. Detailed post hoc analyses are provided in the Appendix A.

The basic (control) protease activity remained at a relatively constant level during the first five days of the experiment, then decreased with the insect age (Figure 11). The main effect of “NP type” was significant, indicating a greater decrease in protease activity in the AgNP-treated groups compared with the control. A significant “concentration” factor confirmed the general inhibitory effect of NPs on protease activity, which was most noticeable in the Ag40 group, followed by the Ag4, GO20, and GO2 groups. The “time” factor also significantly influenced this enzyme’s activity, particularly evidenced by its decrease in the control and NP-treated crickets. This may suggest the diminishing importance of protein digestion in aging insects. However, significant interactions between “time”, “NP type”, and “concentration” factors indicated a slightly different course of changes in protease activity over time, which varied across all concentrations of both tested nanoparticles (Figure 11 and Appendix A and Table 4, Appendix A).

The lipase activity in the control group remained very low and almost constant throughout the experiment. Adding nanoparticles to the cricket diet increased the activity of this enzyme. The “NP type” effect was significant (Table 4 and Appendix A) and indicated stronger enzyme stimulation by AgNPs. This stimulation was dose-dependent, and the “concentration” factor, tested separately for both NPs, showed a stronger positive dependence of the enzyme activity on AgNPs than GO concentration. The observed increase in lipase activity on days 16 and 21 in NP-treated groups was confirmed by a significant “time” effect. The observed effects suggested a potentially increasing importance of fat digestion with the age of the insects when exposed to AgNPs. GO administered at the lowest concentration hardly caused any changes in lipase activity, but other applied concentrations of GO resulted in similar stimulation of lipase activity (Figure 12 and Appendix A and Table 4, Appendix A).

## 4. Discussion

The results obtained in this study indicate that graphene oxide (GO) in the range 2–200 ppm did not result in a remarkable impairment of body weight in *A. domesticus*, except for the GO200 group in which a transient decrease in body weight during the initial period of the experiment was observed (Figure 2). Greater food assimilation in this group compensated for this effect (Figure 6). This result is interesting because exposed adults have a limited increase in body weight and mainly during the first few days after the final molting. So, this measure aimed to capture any potential adverse effects of NPs at the organismal level as a consequence of physiological and biochemical disturbances. Interestingly, GO did not decrease gut cell viability, and we noticed the opposite—a significantly reduced percentage of dead cells in the GO2 and GO20 groups on day 21 (Figure 6). Hence, we confirmed results from our previous, more comprehensive studies, showing a slight decrease in the percentage of dead cells in crickets exposed to 200 μg GO/g of food. However, simultaneously, a significant increase in the percentage of apoptotic cells, cells with intensified oxidative stress (ROS+), and DNA damage occurred. Additionally, histological examination revealed moderate adverse changes in digestive cells but only in the midgut. Moreover, these changes did not affect regenerative cells [42]. The observed increase in food assimilation in the GO200 group or the decrease in the percentage of dead cells suggested a compensatory mechanism that could alleviate initial adverse changes in the body weight of exposed crickets. However, it cannot be ruled out that this compensatory effort may involve tradeoffs at the expense of other functions, like reproduction, physical activity, or life span. These aspects could be investigated in future studies.

AgNPs had a noticeable effect on body mass, food consumption, and assimilation, especially in the Ag40 group, and these results correspond to previous studies. For example, a reduction in body mass resulting from exposure to AgNPs at the highest applied concentration, 25 mg/L, was observed in *D. melanogaster* [43]. Yasur and Pathipati [13] applied AgNPs in the concentration range of 500 to 4000 µg/mL to lepidopteran pests, *Spodoptera litura* and *Achaea janata,* and noticed a decrease in their body weight proportional to the increase in AgNP concentration. Furthermore, the study revealed that while low doses of AgNPs accumulated in the gut, the majority were excreted in the feces. Also, in the beetle *Blaps polychresta*, a dose of 30 ppm per body weight was sufficient to induce numerous abnormalities in midgut cells [28]. In our study, we did not observe a linear relationship of body weight, consumption, and assimilation change with the concentration of AgNPs in the food, but a notable decrease in body weight in the Ag40 group, concomitant with decreased food consumption and assimilation, remains an intriguing observation. This outcome raises the question of whether AgNPs can alter the quality, nutritional value, or even the taste of food. It is likely that at a concentration of 40 ppm, AgNPs do not form large agglomerates and clusters. However, at a concentration of 400 ppm, such structures may already appear, limiting the total surface area of ion release, the effects of which may be potentially unfavorable. Undoubtedly, such a finding necessitates further investigation.

When comparing the effects of both types of nanoparticles, it is essential to remember the differences in their structure and, consequently, the mechanisms by which they interact with cells. AgNPs may act with a delay as Ag ions are released slowly from the nanoparticle surface over an extended period [26,27,28]. The toxicity of GO is related to the flake size, the degree of oxidation, and the type of surface and edge functionalization [29,44,45].

In the present experiment, we observed a series of changes in the activity of digestive enzymes. These changes were, again, more pronounced and evident mainly in the case of AgNPs (Figure 7, Figure 8, Figure 9, Figure 10, Figure 11 and Figure 12). The alterations in digestive enzyme activity under the influence of NPs, especially in insects, remain highly underexplored. The available results are limited and primarily pertain to AgNPs. Nonetheless, it is documented that in *Spodoptera litura* exposed to AgNPs at an LC50 equivalent concentration, the activities of crucial gut enzymes, such as amylase, protease, lipase, and invertase, were significantly lowered. Furthermore, a clear correlation was observed between the concentration of AgNPs and enzyme activity: the higher the nanoparticle concentration, the more pronounced the inhibitory effect. The activities of amylase and invertase were the most significantly reduced [21]. Our study did not reveal such a clear relationship. However, the activity of digestive enzymes in *A. domesticus* depended on “NP type”, “concentration”, and “time” and the interactions of these factors (Table 4 and Appendix A). It is worth noting that Bharani and Namasivayam [21] used a high AgNP dose ranging from 10^3^ to 10^6^ µg AgNPs/mL, three orders of magnitude higher than in our study. The concentration of NPs we used caused subtle changes in enzymes’ activity that may have caused compensatory reactions, which overlapped with the natural, physiological changes in enzymatic activity occurring throughout the insect’s life span. However, within the concentration range applied in our experiment, one cannot infer a distinct inhibitory effect on digestive enzyme activity (except for protease). On the contrary, the observed stimulation of enzyme activity in most groups and enzymes evokes the notion of hormetic effects aimed at enhancing organism survival under moderate stress conditions [19,46,47,48].

In studies on *Spodoptera litura* and *Achaea janata* exposed to AgNPs, a species-dependent response in β-Glu activity was observed. While an increase in the enzyme’s activity was noted in all experimental groups of *S. litura*, *A. janata* showed a reduction in β-Glu activity in almost all groups compared with the control. The exception was the group treated with 2000 µg/mL, where β-Glu activity intensified compared with the control [13]. In another experiment, the influence of AgNPs on the activity of the gut protease was examined in insecticide-resistant *Helicoverpa armigera* caterpillars. A significant reduction in insect body mass and survival was observed, along with the inhibition of protease activity at higher concentrations, and this effect was dose dependent. Further research by the authors demonstrated that AgNPs can bind with high affinity to the protease at higher concentrations, causing its immobilization and inactivity [22]. Indeed, in our study, we observed the inhibition of protease across the entire range of AgNPs and GO concentrations. Additionally, this effect intensified with the crickets’ age, indicating that not only the nanoparticle–enzyme orientation but also the age of the insects was essential. This suggests that the quantity of synthesized/available enzymes, as well as the ratio between NPs, enzymes, and substrates, plays a significant role. The effect of AgNPs on digestive enzymes has also been studied in vertebrates. In common carp fed a diet containing AgNPs at concentrations of 0.05 to 0.15 µg/g for 60 days, a significant reduction in protease and lipase activity was observed, while amylase activity increased [49]. On the other hand, the exposure of another fish species, butterfly splitfin (*Ameca splendens*), to AgNPs at low concentrations (0.01 to 1.0 µg/mL) lasting 42 days did not change the activity of acid and alkaline phosphatase, amylase, lipase, and trypsin, although this exposure was sufficient to induce morphological changes in tissues and disrupt liver and gonad functions [17].

To the best of our knowledge, except for our preliminary study [19], there are no articles describing the effects of GO on digestive enzymes in insects. There are several studies on potential mechanisms of action of other nanoparticles, also in species other than insects. In general, the type and structural properties of NPs determine their mode of action and shape the effect—inhibition or stimulation of digestive enzyme activity [20,23,50]. When administered at low concentrations, nanoparticles composed of essential metals may elicit a stimulatory effect on digestive enzyme activity. Conversely, at elevated concentrations, they tend to inhibit enzyme activity [50]. Wang et al. [20] proposed three interesting hypotheses to explain the impairment of digestive functions following exposure to NPs. The first assumes a direct impact of NPs on digestive enzymes or their synthesis. The second posits an indirect impact of NPs on digestive enzymes through changes in feeding behavior and feeding activity. The third one focuses on the quality of food, which may change after contamination with NPs, indirectly impairing digestive functions. Based on collected evidence, the authors advocated for a direct influence of nanoparticles on enzyme activity or their synthesis [20]. Recent studies by Muhammad et al. [15] on *Bombyx mori* exposed to copper oxide or zinc oxide nanoparticles (CuONPs or ZnONPs) resulted in reduced gene expression of α-amylase or both α-amylase and lipase, respectively, that led to reduced α-amylase activity. However, ZnONPs also affected the microbiome, decreasing species richness and diversity and increasing the abundance of pathobionts. Consequently, reduced survival and cocoon production were observed [15].

A slightly different approach to studying metal NPs involves investigating their modulatory effect on the activity of digestive enzymes, focusing on the structures formed during laboratory-based synthesis [51,52,53]. These studies help unravel the mechanisms of enzyme–NP conjugate formations, hence the role of NPs in modulating enzyme activity [51,53,54,55]. For instance, citrate-stabilized gold nanoparticles (cit-AuNPs) modulate α-amylase activity, with the highest increase observed at the lowest concentration, forming enzyme–NP complexes that positively influence the active site orientation. However, higher NP concentrations reduce enzyme activity due to larger agglomerates [52]. Similar findings show increased enzyme activity when immobilized on gold or biosynthesized silver nanoparticles [53]. Metal NPs can also impair the catalytic functions of other digestive enzymes [51,54,55]. Thus, NPs can act as catalysts within specific concentration ranges, facilitating enzyme–substrate interactions. A separate issue requiring further research is the possibility of binding food component molecules delivered with ingested food on the surface of NPs. The formation of such structures could theoretically influence the retention time of specific nutrients in the gut and its accessibility to digestive enzymes.

## 5. Conclusions

In summary, the subtle changes induced by GO or AgNPs at the used concentrations appear to be safe for *Acheta domesticus*. Notably, AgNPs exerted a more evident influence on insect body weight and consumption than GO. The observed non-linear relationship between AgNP concentration and body weight, consumption, and assimilation may result from possible agglomerate formation that affects the NPs reactivity. However, further detailed studies are necessary to confirm this suggestion. Generally, NPs at the used concentrations stimulated digestive enzymes’ activity, except for protease, resembling a hormetic effect. Whether these observed changes are a tradeoff in a struggle to survive the sublethal but prolonged stress caused by NPs remains a question. Future studies should consider these possible and postponed consequences like the reproductive fitness and life span of the exposed organisms.

## Data Availability

Raw data are provided on the RepOD database (accession: doi:10.18150/SLYEJH).

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
