# Peer review of "Concentration- and Time-Dependent Dietary Exposure to Graphene Oxide and Silver Nanoparticles: Effects on Food Consumption and Assimilation, Digestive Enzyme Activities, and Body Mass in Acheta domesticus"

_insects, 2024, doi:10.3390/insects15020089_

Round 1

Reviewer 1 Report (Previous Reviewer 1)

Comments and Suggestions for Authors

The authors have reanalyzed their results and have presented a greatly improved manuscript.

However, the manuscript contains errors and omissions that need to be corrected before it is suitable for publication.

I have written suggestions and numbered points on a scanned copy of the manuscript.

Numbered points (see scanned manuscript)

1. L205 - Should be µmol rather than µm, I presume.

2. Tables 1, 2, and 3, and 4. F values can only be understood if provided with the treatment and error degrees of freedom (both df1 and df2). Could you indicate df1, df2 in the Table or as a footnote?

3. Excessively long p values should be given to three decimal places as they have been in Table 4.

4. If you are indicating NO sig. difference, p should be >0.05, not p<0.05.

5. Table 5. What is the point of showing analysis of CFC and CFA together? You do not mention this in the text. Are you trying to indicate something here? Does this have a clear biological interpretation?

6. Confused text. Is a figure missing here?

7. Figure 5. I found it impossible to understand this labeling system. What are you trying to show with this combination of letters and symbols? Please simplify it or explain it clearly.

8. The penultimate sentence of the Conclusions needs rewriting.

There are formatting issues in the References section.

Comments on the Quality of English Language

Needs moderate editing during journal production.

Author Response

As in the attachment.

Reviewer 2 Report (New Reviewer)

Comments and Suggestions for Authors

The manuscript explores the impact of graphene oxide (GO) and silver nanoparticles (AgNPs) on the digestive functions of Acheta domesticus, shedding light on the physiological responses of this insect model to nanoparticle exposure. The study presents valuable insights into the intricate interactions between nanoparticles and insect digestive systems, laying the groundwork for further research in this evolving field.

The overall content is engaging, and I am inclined to support its publication. However, attention is required for minor language and structural issues. There are instances of repeated interpretations and redundant lines that may lead to confusion and potential rejection. It is essential to meticulously reorganize and edit the manuscript for clarity.

Specifically, the incomplete sentence at Line 123 and the repeated caption for "Box 1" need addressing. Furthermore, the interpretation for Figure 2, Figure 3, and Figure 4 (Lines 314-320) appears confusing; clarification or potential removal from the text should be considered.

Lastly, instances of repeated interpretations at Line 359 and Line 420 need to be rectified to enhance overall coherence.

Once these issues are diligently addressed, I recommend accepting the manuscript for publication.

Comments on the Quality of English Language

The English proficiency in the manuscript is generally good, with well-structured sentences and appropriate use of scientific terminology. The writing is clear and concise, facilitating the communication of complex scientific concepts. However, there are a few instances where sentence structure and wording could be refined for enhanced clarity. Additionally, some paragraphs are lengthy and could benefit from breaking down into smaller, more digestible segments to improve readability.

For example, in the introduction, the sentence starting with "Most studies highlighting innovative applications..." is quite long and may be challenging for readers to follow. Breaking it into shorter sentences or using more punctuation for clarity could improve comprehension.

In the results and discussion sections, maintaining a balance between detail and conciseness is crucial. Some paragraphs, such as those discussing the effects of nanoparticles on body weight and digestive enzymes, could be structured more clearly for ease of understanding.

Overall, while the manuscript demonstrates a solid command of English, minor revisions for sentence structure, paragraph organization, and clarity could enhance the overall readability and impact of the paper.

Author Response

As in the attachment.

Round 2

Reviewer 1 Report (Previous Reviewer 1)

Comments and Suggestions for Authors

The authors have addressed my previous concerns. The manuscript is suitable for publication in my opinion.

Comments on the Quality of English Language

Minor editing required during journal production.

This manuscript is a resubmission of an earlier submission. The following is a list of the peer review reports and author responses from that submission.

Round 1

Reviewer 1 Report

Comments and Suggestions for Authors

Alian et al report the results of a study on the effects of silver and graphene oxide nanoparticles (NPs) on the feeding behavior, growth and intestinal enzymes of the house cricket A. domesticus.
The manuscript requires major improvement and reanalysis of the data in my opinion.
MAJOR POINTS
(I) The manuscript is very long and wordy and reads like a student thesis.
(II) There are numerous misleading statements on trends that have limited statistical support but are reported as general facts. This needs correction throughout the manuscript.
(III) The experimental design of Fig 2, Fig 4, is one of repeated measures in which groups of insects were sampled repeatedly over time. The appropriate analysis is therefore one of repeated measures ANOVA or a mixed-effects model with time. This would allow us to identify the significance of the MAIN effects of NP concentration, time and concentration × time interaction. At the present it appears that the authors have focused on transitory effects that appear and disappear over time by applying numerous one-way ANOVAs at each time point. The biological significant of transitory effects is not particularly interesting or biologically relevant compared to the main effects.
(IV) The experimental design of Fig 7, 8, 9, 10, 11, 12 involves insects sampled destructively at different time points to measure enzymatic activities. Again the authors focus on on transitory effects that appear and disappear over time by applying numerous one-way ANOVAs at each time point. I think the authors should fit mixed-effects models to determine the main effects of the different NP concentrations over time. The mixed models would be very suitable for this type of data and more informative than the use of multiple ANOVAS especially as some of the data appear to have issues of heteroscedasticity.
I have written numbered points on a scanned copy of the manuscript
Numbered points.
1. Stimulatory effects? But this was variable and only applied to certain enzymes, correct? Is it misleading to state that low concentrations of NPs had stimulatory effects when this was only statistically supported under certain conditions?
2. What do these codes mean? Not explained.
3. (lines 50-65) This is a long preamble on the presence of NPs in food – please focus and summarize.
4. Please support this assertion with evidence.
5. This long text on hypotheses being tested looks like it was taken from a student thesis. Scientific articles do not present hypotheses in this way. You have already stated the aim of the study. Delete this text.
6. Retain short text on cricket biology – this is fine.
7. Delete additional text on cricket biology from the Methods section.
8. I was unable to determine the composition of "Kanisan Q" because all the information I Googled was in Polish. As diet is a MAJOR aspect of this study, please provide information on the composition of this diet (apparently designed to feed rabbits?). You could place nutritional/bromatological information in Supplemental Material, perhaps?
9. Crickets do not have larvae. They have nymphs.
10. Just to clarify – please mention that each treatment comprised SIX replicates, correct?
11. Please explain that assimilation was calculated as gain in mass (dry wt) per day.
12. Did you count the number of dead cells or the percentage of dead cells? (i.e. dead/total cells)?
13. How does the cell counter distinguish between living and dead cells? Are dead cells stained?
14. What volume of buffer was used?
15. Did any insects die in the treatments or the control? Was there any lethal effect of NPs?
16. You state that there was a biological effect (slight stimulation of consumption) in the first half of the experiment, but then say that this was not significant. So really, there was NO effect, and it is misleading to infer that there was one.
You then repeat this inference referring to an effect in the 40 ug treatment that was only significant at a single timepoint (11-16 days). This is misleading in my opinion.
17. You refer to an effect in the second half of the experiment, but in fact the effect is only significant at the last two timepoints of the experiment.
18. Referring to Fig 3, you state that the 40 µg AgNP treatment significantly inhibited food consumption, as a general effect, whereas it appears that these cumulative data were not subjected to statistical analysis, correct?.
19. Referring to Fig 3 you state that there was an effect in the second half of the experiment, but the effect only appeared at the last two timepoints.
20. I had doubts as to whether there was a real effect here or an overstatement given that the cumulative data were not analyzed statistically.
21. Referring to Fig 5 – can you say that there was a significant difference if cumulative assimilation data were not subjected to analysis (or so it seems)?
22. Again you emphasize a reduction in dead cells, but this was only significant for AgNPs.
23. The data in Fig 6 appear to have issues of heteroscedasticity. Did you test these data with Levene's test prior to ANOVA?
24. You state that GO intensified amylase activity as a general statement. Then you say that this only was significant at the highest dose, which makes the general statement rather misleading. (also, should be "concentration" as 200 µg/g is a concentration not a dose).
25. Again the data in Fig 7 appear to have issues of heteroscedasticity. Please check with Levene's test. Explain why only half the error bars are shown.
26. What was the main effect of NP concentration – was the main effect significant? Transient changes are unlikely to be biologically relevant, but overall (main) effects are more interesting.
27. Fig 8. Please indicate significance of the main effects of time, NP concentration and their likely interaction.
28. This is a long detailed explanation of transient differences that are unlikely to be biologically important.
29. Fig 9. Please indicate significance of the main effects of time, NP concentration and their likely interaction.
30. Again you make a general statement that was only supported statistically at certain timepoints. This is misleading.
31. Same issue.
32. Fig 10. Fig 1, Fig 12. Please indicate significance of the main effects of time, NP concentration and their likely interaction.
33. Again, this is an overstatement of an effect that was only significant at two timepoints.
34. This is a paragraph of preamble that was already stated in the Introduction. Delete.
35. You are reiterating results here.
36. To have calculated the LC50 the NPs must have been rather toxic or high concentrations used to treat insects, so how relevant are these findings on S. litura compared to the 'usual' exposure that a caterpillar would receive?
37. 10e3 – 10e6 µg/mL is a concentration, not a dose.
38. This is a long and detailed description of studies in other organisms that reads like a student thesis. Please summarize general trends observed in other species.
39. This is a very long text on enzymes conjugated to NPs, that is not relevant to this study as conjugated enzymes were not used in your study. Delete.
40. The conclusions should be a brief 5 or 6 lines of text highlighting the principal conclusion and the principal next steps that follow on from this study.

Comments on the Quality of English Language

Some editing required, especially in the summary and abstract.

Author Response

As in the attachment

Reviewer 2 Report

Comments and Suggestions for Authors

I would like to appreciate a tremendous amount of work, evident from the numerous experiments conducted, that was made by authors of this manuscript. However, the inconsistency in the obtained results leaves me wondering about their contribution to development of existing knowledge. For instance, in the case of AgNPs, there's an absence of a linear relationship in the crickets' response to dosage increments, which is intriguing. Furthermore, only in the Ag40 group the body mass was lower than in the control group. When focusing solely on Ag40, the authors noticed a decrease in food consumption and assimilation, yet the number of dead cells in the intestines remains unchanged. This suggests that the decrease in body mass might not be linked to the extent of intestinal damage. Could it then be related to enzyme activity? If so, the results for Ag40 would significantly differ from other AgNP doses. However, here too, there is inconsistency. In most cases, the effect on enzyme activity is similar across different AgNP concentrations. The only exception is β-glucosidase, where the effect after administering Ag40 differs from Ag4 and Ag400.

This issue also highlights how the hypotheses were presented: "Consumption of food contaminated with nanoparticles (GO, AgNPs) in various concentrations will not affect or increase or decrease selected enzymes." The results obtained are so inconclusive that drawing an overarching conclusion becomes challenging, as evident in the Conclusions. Partial acceptance of one hypothesis and a different one for another enzyme leads to a situation where research hypotheses are not properly formulated or verified. In summary, I believe the work contains a plethora of results that lack coherence. The authors should analyze the results and rethink what they aim to convey to the readers.

In their discussion, the authors themselves admit, "Our study shows that the admixture of AgNPs or GO, at least in the concentrations covered by our research, had a limited impact on the amount of food consumption and its assimilation, especially in the case of GO." Therefore, perhaps changes in enzyme activity, observed after AgNPs treatment, might not significantly impact the functioning of the examined insects. In other words, there is some change in physiology, but it does not significantly affect the life of crickets.

Minor comments:

The English should be improved throughout for clarity

Line 45: the sentence should be clarified – did the authors meant exposed organisms

Hypotheses:

Never have I encountered a situation where the formulated research hypotheses are contradictory. We propose one research hypothesis and either confirm it or not

One research hypothesis should be formulated. It can be developed into subpoints.

Line 308

‘After 21 days, insects from the Ag40 group still had a lower average body mass gain than the control group, although the difference was no longer statistically significant’

If the difference is not statistically significant, then there is no difference

Line 607

dose ranging from 1000 to 1 × 106 µg AgNPs/mL.  – I have difficulty to understand what is the exact dose?

Author Response

As in the attachment
